# ProCLIP: Progressive Vision-Language Alignment via LLM-based Embedder

## Abstract

The original CLIP text encoder is limited by a maximum input length of 77 tokens, which hampers its ability to effectively process long texts and perform fine-grained semantic understanding. In addition, the CLIP text encoder lacks support for multilingual inputs. All these limitations significantly restrict its applicability across a broader range of tasks. Recent studies have attempted to replace the CLIP text encoder with an LLM-based embedder to enhance its ability in processing long texts, multilingual understanding, and fine-grained semantic comprehension. However, because the representation spaces of LLMs and the vision-language space of CLIP are pretrained independently without alignment priors, direct alignment using contrastive learning can disrupt the intrinsic vision-language alignment in the CLIP image encoder, leading to an underutilization of the knowledge acquired during pre-training. To address this challenge, we propose **ProCLIP**, a curriculum learning-based progressive vision-language alignment framework to effectively align the CLIP image encoder with an LLM-based embedder. Specifically, ProCLIP first distills knowledge from CLIP's text encoder into the LLM-based embedder to leverage CLIP's rich pretrained knowledge while establishing initial alignment between the LLM embedder and CLIP image encoder. Subsequently, ProCLIP further aligns the CLIP image encoder with the LLM-based embedder through image-text contrastive tuning, employing self-distillation regularization to avoid overfitting. To achieve a more effective alignment, instance semantic alignment loss and embedding structure alignment loss are employed during representation inheritance and contrastive tuning. Extensive experiments show ProCLIP achieves 6.8% to 13.5% improvement on zero-shot classification and presents excellent performance on cross-modal retrieval, multilingual cross-modal retrieval, and fine-grained understanding tasks, demonstrating the effectiveness and robustness of ProCLIP. To support reproducibility and facilitate further research, we will release the training code and model weights.

## 1 Introduction

CLIP demonstrates remarkable zero-shot recognition capabilities by learning joint vision-language representations through contrastive learning on large-scale image–text pairs (Radford et al., 2021). Serving as a bridge between vision and language, it is widely adopted in multiple downstream tasks such as image–text retrieval (Yang et al., 2023), text-to-image generation (Wang et al., 2022), and open-vocabulary object detection (Wu et al., 2023b). However, the original CLIP model relies on English text captions with a maximum length of 77 tokens as its supervisory signal (Zhang et al., 2024). This design limits its capacity to process long-form text and restricts input to English-only (Tschannen et al., 2025). Additionally, due to the absence of supervision for fine-grained textual semantics, this limitation further impedes its semantic understanding capability (Hu et al., 2025).

To overcome these limitations, methods such as Long-CLIP (Zhang et al., 2024) interpolate positional embeddings and fine-tune on long text–image pairs to extend the input length. While effective for long-text understanding, such approaches still fall short in enhancing CLIP's fine-grained semantic understanding and multilingual capabilities. Recently, LLM exhibits remarkable proficiency in natural language processing, and it has pivoted towards harnessing decoder-only architectures for

effective representation learning (BehnamGhader et al., 2024; Lee et al., 2024). Following this trend, methods such as FLAME (Cao et al., 2025) and LLM2CLIP (Huang et al., 2024) propose to replace CLIP's original text encoder with LLM-based embedders. By leveraging the rich open-world knowledge inherent in LLMs, these approaches aim to enhance CLIP's representational capacity—particularly in processing longer and more complex image captions. However, these methods align the CLIP image encoder directly with the LLM-based text embedder through contrastive learning, while neglecting the rich pretrained knowledge within CLIP. This "*from-scratch alignment*" compels both encoders to learn a new representation space from scratch, disregarding the original CLIP alignment knowledge. Such an approach increases the risk of overfitting, particularly when training data is scarce, thereby compromising model generalization. This observation leads to a critical research question: *How can we systematically leverage CLIP's pretrained knowledge to achieve efficient cross-modal alignment with an LLM-based embedder while preserving generalization capability?*

In this paper, we propose **ProCLIP**, a simple yet effective progressive vision-language alignment framework enhancing the CLIP. ProCLIP leverages curriculum learning to first guide the LLM-based embedder (only MLP trainable) to adapt to the CLIP text encoder's representation space, and then uses contrastive learning to further learn joint image-text representations. Specifically, ProCLIP first distills knowledge from the original CLIP text encoder into the LLM-based embedder, establishing an initial alignment between the CLIP image encoder and LLM-based embedder. Subsequently, we conduct contrastive learning on image–text pairs to further improve this alignment. Since the LLM-based embedder is already partially aligned with the CLIP image encoder during the prior stage, the contrastive optimization process becomes more stable and preserves generalization more effectively. To further mitigate overfitting, we impose a self-distillation constraint on the CLIP image encoder throughout this stage, which stabilizes training and improves generalization. To prove the effectiveness of ProCLIP, we evaluate it on multiple tasks across diverse data scales and model sizes. Extensive experiment results demonstrate that ProCLIP achieves consistently significant improvements. The main contributions of this paper are summarized as follows:

- We **highlight the limitation of previous works**: previous methods fail to fully exploit the pretrained knowledge in CLIP, and their reliance on simplistic contrastive learning for cross-modal alignment significantly compromises CLIP's inherent generalization capabilities.

- We **propose ProCLIP , a simple but effective Progressive vision-language alignment framework to enhance CLIP**. ProCLIP initially distills the pretrained knowledge into the LLM-based embedder. After that, ProCLIP utilizes contrastive fine-tuning constrained by self-distillation to further enhance cross-modal alignment while preserving the model's inherent generalization capacity.

- We **conduct extensive experiments on multiple tasks across diverse data scales and model sizes**. Compared to the baseline, ProCLIP achieves 6.8% to 13.5% improvement on zero-shot classification and performs strongly on other tasks, including short-text cross-modal retrieval, long-text cross-modal retrieval, multilingual cross-modal retrieval, and fine-grained understanding.

## 2 RELATED WORK

**Vision-Language Contrastive Learning.** Vision-language contrastive learning aims to learn robust multimodal representations by pretraining on large-scale image-text pairs. A seminal work in this area, CLIP (Radford et al., 2021) aligns visual and linguistic representations through contrastive learning, bridging both modalities in a shared semantic space. As a bridge between vision and language, CLIP has been widely applied in multimodal learning. It enables a variety of natural language-guided open-vocabulary recognition tasks, including image classification (Zhou et al., 2022b;a; Kim et al., 2024), open-vocabulary semantic segmentation (Ding et al., 2022; Li et al., 2022; Ghiasi et al., 2022; Xu et al., 2022; Cho et al., 2024; Lan et al., 2024), and open-vocabulary object detection (Du et al., 2022; Kaul et al., 2023). However, CLIP remains fundamentally constrained by its text encoder's limited capacity and fixed input length, which hinders its ability to process multilingual and long texts and model fine-grained semantics. To mitigate these issues, several methods have been introduced. Long-CLIP (Zhang et al., 2024) extends the input length via positional embedding interpolation, yet still fails to capture nuanced semantic relationships. LoTLIP (Wu et al.,

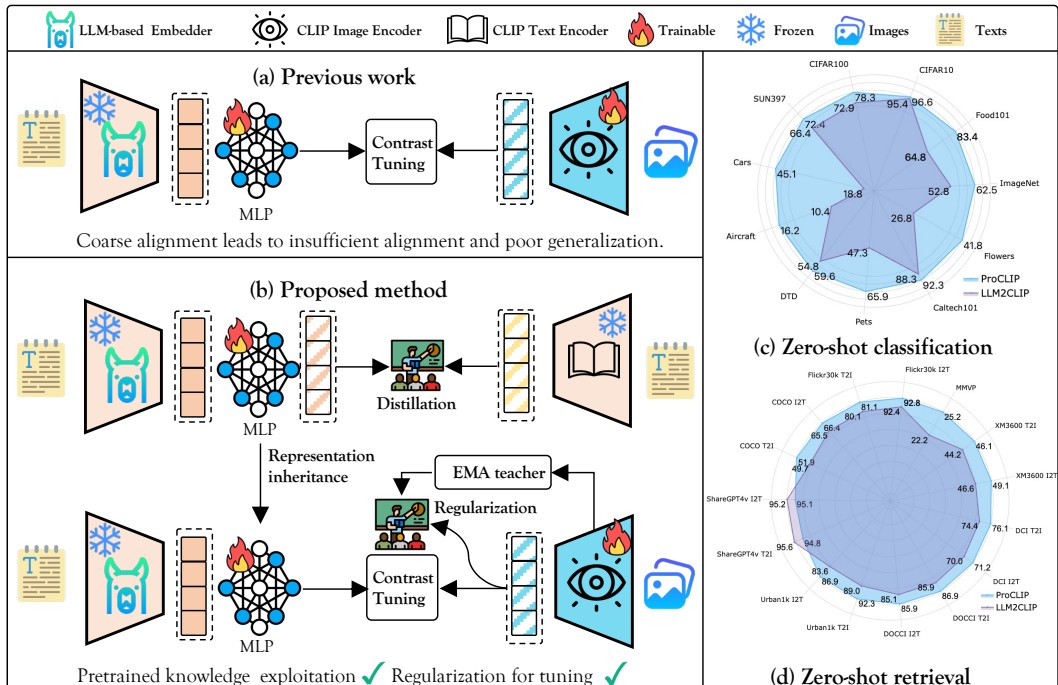

Figure 1: Previous work directly aligns the LLM-based embedder with the CLIP image encoder, disregarding the valuable knowledge embedded in the pre-trained CLIP model. In contrast, ProCLIP first transfers knowledge from CLIP's text encoder to the LLM embedder via distillation, establishing an initial alignment. It then refines the alignment between the CLIP image encoder and the LLM-based embedder through image-text contrastive learning regularized by self-distillation.

2024) incorporates corner tokens to aggregate diverse textual information, preserving short-text understanding while significantly improving performance on long texts. Nevertheless, constrained by the capabilities of the text encoder, LoTLIP cannot incorporate additional open-world knowledge and remains unable to handle multilingual inputs.

**LLMs for Representation Learning.** Large language models have presented remarkable proficiency across a wide range of natural language processing tasks Touvron et al. (2023); Achiam et al. (2023); Bai et al. (2023); Liu et al. (2024). Recent research has pivoted towards harnessing decoder-only architectures for effective representation learning. For instance, LLM2Vec (BehnamGhader et al., 2024) converts pre-trained decoder-only LLMs into versatile text encoders by incorporating three principal advancements: bidirectional attention mechanisms, masked next-token prediction, and unsupervised contrastive alignment. Meanwhile, Qwen3-Embedding (Zhang et al., 2025b) capitalizes on the Qwen3 model's strong multilingual understanding and generation abilities. By integrating a large-scale unsupervised pretraining and supervised fine-tuning on high-quality data, it achieves state-of-the-art performance on the MTEB benchmark (Muennighoff et al., 2022). Inspired by these advances, recent works (Huang et al., 2024; Cao et al., 2025; Zhang et al., 2025a) attempt to enhance CLIP by replacing its text encoder with a powerful LLM-based embedder, thereby improving its ability to process multilingual, longer, and more complex textual inputs. Although these approaches present promise, their alignment strategies remain overly coarse and often lead to degraded generalization. Developing more refined and effective alignment techniques thus remains a critical and open research challenge.

**Knowledge Distillation.** Knowledge distillation Hinton et al. (2015) is widely used in deep learning to enhance model performance and reduce computational complexity. Typically, a larger teacher model transfers knowledge to a smaller student model by guiding the learning of features or output distributions. Alternatively, self-distillation methods enable knowledge transfer within a single model, where deeper layers supervise shallower ones Zhang et al. (2019). In the context of CLIP, several distillation techniques have been introduced. TinyCLIP Wu et al. (2023a) employs affinity mimicking to capture cross-modal interactions during distillation, allowing the student to replicate the teacher's alignment behavior in a shared affinity space. CLIP-KD Yang et al. (2024a)

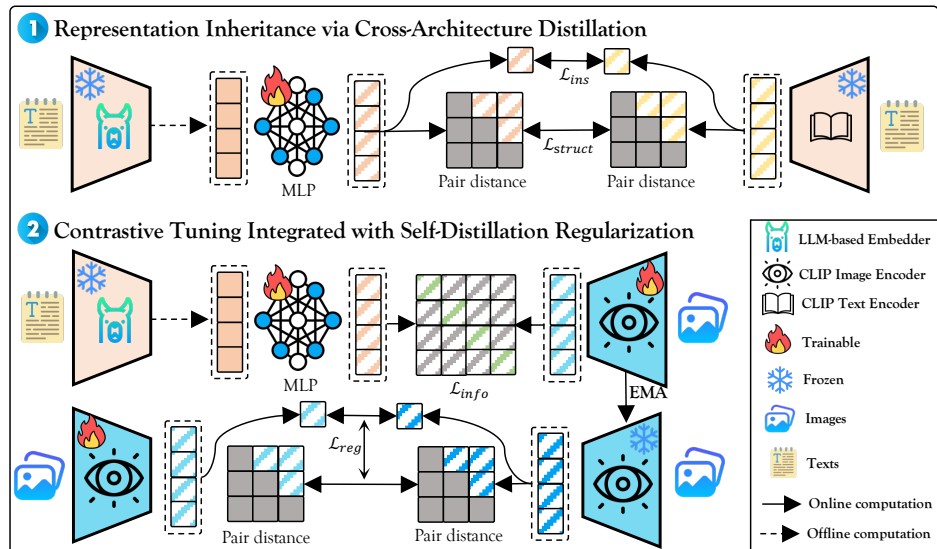

Figure 2: The training pipeline of our proposed ProCLIP. It consists of representation inheritance via cross-architecture distillation and contrastive tuning integrated with self-distillation regularization.

integrates multiple strategies—including relation-based, feature-based, gradient-based, and contrastive distillation—to maximize feature similarity between the teacher and student models. CLIP-CID Yang et al. (2024b) utilizes cluster-instance discrimination to transfer semantic knowledge from the teacher, enabling the student to develop a richer understanding of the pretraining data. Different from the above method, this paper introduces a self-distillation mechanism to mitigate catastrophic forgetting during training and preserve the generalization capabilities of the model.

## 3 METHODOLOGY

In this section, we first introduce the preliminary (Sec. 3.1), including contrastive language-image pre-training and improving CLIP with an LLM-based embedder. Then we present our proposed ProCLIP framework, which comprises two primary training stages: 1) Representation Inheritance via Cross-Architecture Distillation (Sec. 3.2). 2) Contrastive Tuning Integrated with Self-Distillation Regularization (Sec. 3.3).

### 3.1 PRELIMINARY

**Contrastive Language-Image Pre-training.** Contrastive Language-Image Pre-training (CLIP) Radford et al. (2021) learns to align images and text from large-scale image–text pairs through contrastive learning, bridging both modalities in a shared embedding space. Given a batch of image-text pairs $\{(I_i, T_i)\}_{i=1}^{\mathcal{B}}$, the image encoder $\mathcal{E}_I$ and text encoder $\mathcal{E}_T$ map them into the joint semantic space as $\{(v_i, t_i)\}_{i=1}^{\mathcal{B}}$. To optimize both encoders in a dual-tower architecture, a symmetric contrastive learning objective is imposed on the resulting representations:

$$\mathcal{L}_{CLIP} = -\sum_{i=1}^{\mathcal{B}} \left[ \underbrace{\log \frac{\exp(v_i \cdot t_i^\top / \tau)}{\sum_{j=1}^{\mathcal{B}} \exp(v_i \cdot t_j^\top / \tau)}}_{\text{text-to-image}} + \underbrace{\log \frac{\exp(t_i \cdot v_i^\top / \tau)}{\sum_{j=1}^{\mathcal{B}} \exp(t_i \cdot v_j^\top / \tau)}}_{\text{image-to-text}} \right]. \quad (1)$$

However, the native CLIP text encoder is limited to sequences of up to 77 tokens. A common solution is to interpolate the position embeddings of the CLIP text encoder and fine-tune the model. Alternatively, one may replace the CLIP text encoder with an LLM-based embedder. The latter approach not only improves long-text understanding but also enhances multilingual understanding and fine-grained semantic comprehension, resulting in a more versatile vision-language dual-encoder. In this work, we investigate a more efficient alignment strategy that leverages an LLM-based embedder to enhancing CLIP's comprehensive capabilities.

**Improving CLIP with LLM-based Embedder.** LLM2CLIP Huang et al. (2024) first introduces an LLM-based embedder into CLIP, demonstrating enhanced long-text understanding. Given an LLM-based encoder $\mathcal{G}_T$, it encodes texts $\{T_i\}_{i=1}^{\mathcal{N}}$ offline into embeddings $\{t_i'\}_{i=1}^{\mathcal{N}}$. This process is typically performed in an offline manner. During contrastive fine-tuning, a multilayer perceptron (MLP) is used to map $\{t'\}_{i=1}^{\mathcal{N}}$ into the CLIP embedding space for dimensional alignment. The mapped text features and the image features from the CLIP image encoder are then optimized via the contrastive loss in Eq. 1, resulting in a newly aligned representation space. However, applying contrastive learning directly to fine-tuning data to optimize the MLP and vision encoder hinders the convergence of the new dual-tower architecture to an optimal parameter space. This arises because the text representations from the LLM-based embedder and MLP lack prior alignment with the vision encoder. Moreover, unconstrained fine-tuning may also cause excessive drift from the original pre-trained representation, while limited fine-tuning data (*e.g.*, 3M samples) cannot compensate for the knowledge acquired during large-scale pre-training (*e.g.*, 400M samples). To overcome these challenges, we propose a progressive alignment pipeline that improves multimodal alignment while preserving pre-trained knowledge.

### 3.2 Stage 1: Representation Inheritance via Cross-Architecture Distillation.

Given a pre-trained image and text encoder of the CLIP model $\{\mathcal{E}_I, \mathcal{E}_T\}$ and a pre-trained LLM-based embedder $\mathcal{G}_T$, our goal is to replace the CLIP text encoder $\mathcal{E}_T$ with the LLM-based embedder $\mathcal{G}_T$ to enhance comprehensive abilities. Consistent with prior works Huang et al. (2024); Cao et al. (2025); Zhang et al. (2025a), we initially extract embeddings from textual captions offline using $\mathcal{G}_T$: $t' = \{\mathcal{G}_T(T_i) \in \mathbb{R}^d\}_{i=1}^{\mathcal{N}}$, where $d$ represents the embedding dimension of the LLM-based embedder.

The embedding space of the LLM-based embedder exhibits no prior alignment with the CLIP image-text representation space. To bridge this gap, we adopt a cross-architecture distillation strategy that transfers knowledge from the CLIP text embedding space to the LLM embedding space. Specifically, given a batch of texts $\{T_i\}_{i=1}^{\mathcal{B}}$, we first utilize a single-layer MLP to unify the dimensions of LLM embeddings and CLIP text embeddings. To facilitate fine-grained semantic alignment, we propose an instance semantic alignment loss, denoted as $\mathcal{L}_{\text{ins}}$. This loss function leverages text-only data to distill knowledge from CLIP's text encoder into the LLM-based embedder, defined as follows:

$$\mathcal{L}_{\text{ins}} = \sum_{i=1}^{\mathcal{B}} \|\text{MLP}(t_i') - \mathcal{E}(T_i)\|_2. \tag{2}$$

Since $\mathcal{L}_{\text{ins}}$ only focuses on instance-level alignment without capturing the global embedding structure, we propose the embedding structure alignment loss $\mathcal{L}_{\text{struct}}$. This loss measures inter-sample distances within a batch in both the CLIP text encoder and LLM-based embedder spaces, and aligns the two globally by minimizing their pairwise distance discrepancy. $\mathcal{L}_{\text{struct}}$ is defined as:

$$\mathcal{L}_{\text{struct}} = \sum_{\substack{i,j=1 \\ i<j}}^{\mathcal{B}} \big| \|\text{MLP}(t_i') - \text{MLP}(t_j')\|_2 - \|\mathcal{E}(T_i) - \mathcal{E}(T_j)\|_2 \big|. \tag{3}$$

The overall loss is the first stage is defined as: $\mathcal{L}_{\text{dis}} = \mathcal{L}_{\text{ins}} + \mathcal{L}_{\text{struct}}$.

### 3.3 Stage 2: Contrastive Tuning Integrated with Self-Distillation Regularization.

After the above phase, the $\text{MLP}(\mathcal{G}_T)$ has already been preliminarily adapted to CLIP's vision-language embedding space, making subsequent fine-tuning with vision-language contrastive learning significantly easier. We utilize the InfoNCE loss (Radford et al., 2021) to better align the image embedding $v_i$ and the projected LLM embedding $t_i^* = \text{MLP}(t_i')$, which can be formulated as:

$$\mathcal{L}_{\text{info}} = -\sum_{i=1}^{\mathcal{B}} \left[ \log \frac{\exp(v_i \cdot t_i^{*\top}/\tau)}{\sum_{j=1}^{\mathcal{B}} \exp(v_i \cdot t_j^{*\top}/\tau)} + \log \frac{\exp(t_i^* \cdot v_i^{\top}/\tau)}{\sum_{j=1}^{\mathcal{B}} \exp(t_i^* \cdot v_j^{\top}/\tau)} \right], \tag{4}$$

where $\tau$ is a learnable temperature parameter. Beyond standard contrastive learning, we impose a self-distillation constraint on the CLIP image encoder to mitigate excessive forgetting of pre-trained

Table 1: Cross-modal retrieval performance Recall@1 on multiple datasets.

| Method | Data | Flickr30k | | COCO | | ShareGPT4V | | Urban-1k | | DOCCI | | DCI | | Avg. | |
|---|---|---|---|---|---|---|---|---|---|---|---|---|---|---|---|
| | | I2T | T2I | I2T | T2I | I2T | T2I | I2T | T2I | I2T | T2I | I2T | T2I | I2T | T2I |
| *Model Architecture: CLIP ViT-B/32* | | | | | | | | | | | | | | | |
| CLIP | 400M | 80.3 | 59.8 | 51.5 | 30.6 | 77.3 | 66.0 | 60.9 | 46.8 | 58.1 | 53.4 | 43.1 | 40.3 | 61.8 | 49.5 |
| LLM2CLIP | 3M | 83.5 | 70.1 | 55.6 | 41.1 | 94.2 | **93.4** | 78.2 | 84.2 | 76.2 | 77.1 | 62.2 | 64.4 | 75.0 | 71.1 |
| ProCLIP | 3M | **86.0** | **73.5** | 57.8 | **43.5** | **94.4** | 92.6 | 80.8 | 85.3 | 78.1 | 79.5 | 65.7 | 68.3 | **77.1(+2.1)** | **73.8(+2.7)** |
| LLM2CLIP | 15M | 86.2 | 72.2 | 58.5 | 43.2 | **95.3** | **94.2** | 80.6 | 85.3 | **79.2** | 80.7 | 64.3 | 67.6 | 77.4 | 73.9 |
| ProCLIP | 15M | **86.6** | **72.6** | **59.0** | **43.5** | 94.5 | 93.9 | 82.2 | 85.3 | 78.4 | 80.6 | 67.1 | 69.2 | **78.0(+0.6)** | **74.2(+0.3)** |
| LLM2CLIP | 30M | 87.8 | 72.4 | 61.1 | 44.3 | 96.7 | **95.9** | 86.6 | 88.8 | **82.9** | 82.9 | 67.9 | 69.5 | 80.5 | 75.7 |
| ProCLIP | 30M | **90.2** | **74.6** | **62.4** | **45.9** | **96.8** | **95.9** | 88.5 | 89.9 | **82.9** | 84.1 | 70.6 | 71.9 | **81.9(+1.4)** | **77.0(+1.3)** |
| *Model Architecture: CLIP ViT-B/16* | | | | | | | | | | | | | | | |
| CLIP | 400M | 82.7 | 63.4 | 53.7 | 33.3 | 76.1 | 68.9 | 67.5 | 53.5 | 66.8 | 57.0 | 45.4 | 43.0 | 65.4 | 45.6 |
| LLM2CLIP | 3M | 88.0 | 75.3 | 60.5 | 44.8 | **94.4** | **94.4** | 80.6 | 86.0 | 81.7 | 82.2 | 67.2 | 69.1 | 78.7 | 75.3 |
| ProCLIP | 3M | **89.4** | **77.6** | **61.7** | **46.8** | 94.3 | 93.3 | 82.9 | 88.1 | 81.0 | 82.5 | 67.3 | 72.0 | **79.4(+0.7)** | **76.7(+1.4)** |
| LLM2CLIP | 15M | 88.9 | 76.6 | 62.4 | 46.5 | **95.0** | **95.2** | 84.5 | 88.4 | **83.8** | 85.1 | 69.3 | 72.4 | 80.7 | 77.3 |
| ProCLIP | 15M | **90.8** | **77.9** | **63.2** | **47.8** | 94.2 | 94.9 | 85.8 | 89.6 | 82.5 | 84.6 | 70.2 | 74.0 | **81.2(+0.5)** | **78.0(+0.7)** |
| LLM2CLIP | 30M | 90.2 | 78.1 | 65.4 | 48.5 | **96.8** | **96.4** | 89.7 | 91.3 | **86.2** | 86.8 | 73.1 | 74.8 | 83.6 | 79.3 |
| ProCLIP | 30M | **92.7** | **79.1** | **67.1** | **49.7** | 96.0 | **96.4** | 90.0 | 93.4 | 85.1 | 87.3 | 73.6 | 76.9 | **84.2(+0.6)** | **80.5(+1.2)** |
| *Model Architecture: CLIP ViT-L/14* | | | | | | | | | | | | | | | |
| CLIP | 400M | 86.6 | 64.6 | 57.2 | 36.4 | 78.0 | 68.7 | 68.4 | 56.0 | 65.8 | 63.1 | 45.4 | 43.9 | 66.9 | 55.5 |
| LLM2CLIP | 3M | 92.4 | 80.1 | 65.5 | 49.7 | **95.2** | **95.6** | 83.6 | 89.0 | 85.1 | 85.9 | 70.0 | 74.4 | 82.0 | 79.1 |
| ProCLIP | 3M | **92.8** | **81.1** | **66.4** | **51.9** | 95.1 | 94.8 | 86.9 | 92.3 | 85.9 | 86.9 | 71.2 | 76.1 | **83.0(+1.0)** | **80.5(+1.4)** |
| LLM2CLIP | 15M | 91.3 | 80.6 | 67.0 | 50.6 | **96.3** | 95.3 | 86.4 | 90.5 | 86.4 | 88.5 | 71.7 | 75.3 | 83.2 | 80.1 |
| ProCLIP | 15M | **93.4** | **81.4** | **67.6** | **52.5** | 96.1 | **95.4** | 88.3 | 92.6 | 86.2 | 88.4 | 74.4 | 76.8 | **84.3(+1.3)** | **81.2(+1.1)** |
| LLM2CLIP | 30M | 93.1 | 81.0 | 68.2 | 52.0 | **97.5** | **97.7** | 92.7 | 93.9 | **88.2** | 89.6 | 74.9 | 78.3 | 85.8 | 82.1 |
| ProCLIP | 30M | **94.5** | **81.6** | **69.3** | **53.2** | 96.8 | 97.0 | 93.0 | 94.4 | 87.5 | 89.8 | 75.9 | 79.5 | **86.2(+0.4)** | **82.6(+0.5)** |
| *Model Architecture: EVA02-CLIP ViT-L/14* | | | | | | | | | | | | | | | |
| EVA02-CLIP | 2B | 88.9 | 76.9 | 63.6 | 46.6 | 84.5 | 79.4 | 72.0 | 69.4 | 72.6 | 74.2 | 43.9 | 45.2 | 70.9 | 65.3 |
| LLM2CLIP | 3M | **93.8** | 81.7 | 66.6 | 51.1 | 96.5 | 95.9 | 84.4 | 92.1 | 86.6 | 88.7 | **73.8** | 76.1 | 83.6 | 80.9 |
| ProCLIP | 3M | 93.0 | **82.6** | **68.6** | **53.4** | **96.6** | **96.0** | 88.4 | 93.2 | 87.0 | 89.7 | 71.8 | 78.4 | **84.2(+0.6)** | **82.2(+1.3)** |

knowledge during adaptation—essential for preserving generalization. On the image encoder side, we apply a regularization loss that is symmetric to the one used in the first stage(Eq. 2, Eq. 3):

$$\mathcal{L}_{\text{reg}} = \sum_{i=1}^{\mathcal{B}} \|\mathcal{E}_I(I_i) - \mathcal{E}_I^*(I_i)\|_2 + \sum_{\substack{i,j=1 \\ i<j}}^{\mathcal{B}} \left| \|\mathcal{E}_I(I_i) - \mathcal{E}_I(I_j)\|_2 - \|\mathcal{E}_I^*(I_i) - \mathcal{E}_I^*(I_j)\|_2 \right|, \quad (5)$$

where $\mathcal{E}_I^*$ denotes the EMA (Exponential Moving Average)-updated image encoder obtained as:

$$\mathcal{E}_I^* = \alpha \mathcal{E}_I^* + (1-\alpha)\mathcal{E}_I, \quad (6)$$

where $\alpha$ controls the update rate of the teacher model parameters. The overall loss function of the contrastive tuning stage is defined as $\mathcal{L}_{\text{tune}} = \mathcal{L}_{\text{info}} + \lambda \mathcal{L}_{\text{reg}}$, where $\lambda$ is a loss weight.

## 4 EXPERIMENTS

### 4.1 EXPERIMENTAL SETUP

**Datasets and Benchmarks.** For the alignment dataset, we use CC3M (Changpinyo et al., 2021), CC12M (Changpinyo et al., 2021), and YFCC15M (Thomee et al., 2016),combined the high-quality captions from DreamLIP Zheng et al. (2024). We conduct experiments with data scales of 3M (CC3M), 15M (CC3M + CC12M), and 30M (CC3M + CC12M + YFCC15M) to explore the effects of data scaling. For the benchmark, we perform zero-shot classification on 11 different classification datasets, robustness evaluations on 5 ImageNet variants, retrieval evaluations on 6 datasets, multilingual cross-modal retrieval evaluation on XM3600 Thapliyal et al. (2022), and fine-grained understanding evaluation on MMVP-VLM Tong et al. (2024). Regarding the model, we employ three OpenAI pre-trained CLIP models, ViT-B/32, ViT-B/16, and ViT-L/14, to investigate the effects of model scaling. Additionally, we conduct experiments with pretrained EVA02-CLIP (Fang et al., 2023) ViT-L/14 to assess the impacts of different model architectures. For the LLM-based embedder, we primarily use LLaMA3-8B-CC consistent with LLM2CLIP Huang et al. (2024).

**Implementation Details.** For the representation inheritance phase, we train for four epochs, followed by another four epochs for contrastive tuning. During training, we employ AdamW (Loshchilov, 2019) as the optimizer, with a learning rate of $1 \times 10^{-5}$ and a weight decay of 0.2. The parameters $\beta_1$ and $\beta_2$ are set to 0.9 and 0.98, respectively. In the first stage, the training batch size is set to 1024, while in the second stage it is increased to 4096. The loss weight $\lambda$ is set at 0.0004. Other training details can be found in the supplementary material.

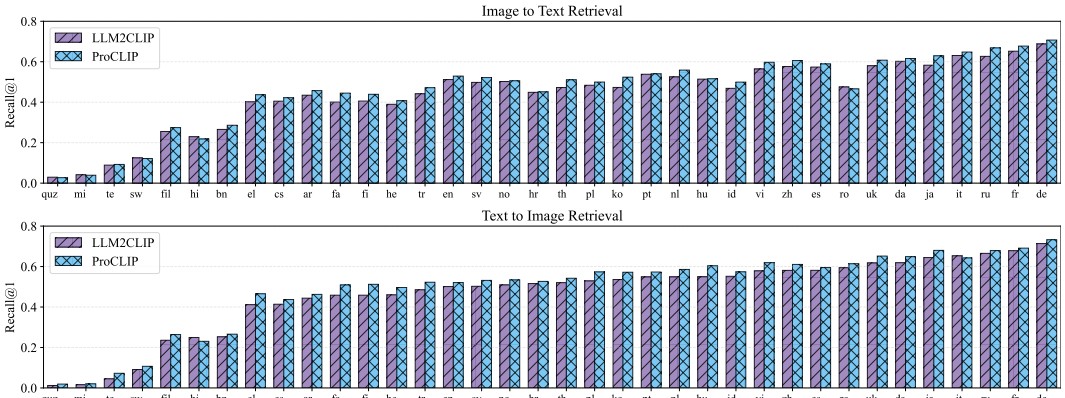

Figure 3: Per-language image-text retrieval performance for LLM2CLIP and ProCLIP on the XM3600 benchmark.

Table 2: Zero-shot classification performance on 11 datasets. The best results are marked in **bold**.

| Method | Data | Food101 | CIFAR10 | CIFAR100 | SUN397 | Cars | Aircraft | DTD | Pets | Caltech101 | Flowers | ImageNet | Avg. |
|---|---|---|---|---|---|---|---|---|---|---|---|---|---|
| *Model Architecture: CLIP ViT-B/32* | | | | | | | | | | | | | |
| CLIP | 400M | 83.1 | 88.7 | 63.5 | 61.5 | 57.6 | 18.8 | 42.8 | 84.6 | 89.4 | 66.0 | 61.9 | 65.2 |
| LLM2CLIP | 3M | 49.6 | 89.2 | 61.5 | 60.3 | 11.5 | 8.6 | 47.8 | 38.0 | 79.0 | 22.6 | 41.0 | 46.3 |
| ProCLIP | 3M | 64.5 | 90.7 | 65.8 | 65.0 | 21.2 | 11.6 | 52.0 | 51.7 | 83.3 | 30.8 | 47.9 | 53.1(+6.8) |
| LLM2CLIP | 15M | 57.2 | 88.3 | 61.4 | 61.3 | 19.6 | 8.4 | 50.6 | 42.3 | 80.7 | 23.5 | 43.3 | 48.8 |
| ProCLIP | 15M | 74.9 | 90.0 | 66.5 | 65.1 | 39.6 | 13.9 | 53.7 | 68.5 | 86.7 | 35.5 | 53.3 | 58.9(+10.1) |
| LLM2CLIP | 30M | 58.5 | 88.3 | 61.0 | 61.2 | 20.6 | 8.4 | 50.3 | 37.6 | 81.7 | 26.0 | 45.1 | 49.0 |
| ProCLIP | 30M | 74.4 | 88.8 | 66.9 | 65.9 | 38.0 | 16.2 | 53.0 | 64.5 | 86.8 | 40.4 | 54.0 | 59.0(+10.0) |
| *Model Architecture: CLIP ViT-B/16* | | | | | | | | | | | | | |
| CLIP | 400M | 87.9 | 89.7 | 66.8 | 63.1 | 63.7 | 22.8 | 45.0 | 87.0 | 90.4 | 67.6 | 67.1 | 68.3 |
| LLM2CLIP | 3M | 56.9 | 92.6 | 64.4 | 62.2 | 15.4 | 11.7 | 50.9 | 46.5 | 82.9 | 23.6 | 45.8 | 50.3 |
| ProCLIP | 3M | 73.1 | 92.5 | 68.9 | 67.9 | 32.3 | 13.5 | 54.1 | 59.8 | 87.0 | 35.8 | 54.8 | 58.2(+7.9) |
| LLM2CLIP | 15M | 63.2 | 90.8 | 64.5 | 62.9 | 27.3 | 9.9 | 52.8 | 50.3 | 83.2 | 23.7 | 46.5 | 52.3 |
| ProCLIP | 15M | 80.3 | 90.8 | 69.7 | 67.4 | 44.3 | 16.5 | 56.7 | 75.8 | 88.4 | 40.8 | 58.6 | 62.7(+10.4) |
| LLM2CLIP | 30M | 64.4 | 90.2 | 64.6 | 63.7 | 27.0 | 11.2 | 55.0 | 45.9 | 84.0 | 27.1 | 49.7 | 53.0 |
| ProCLIP | 30M | 81.0 | 89.3 | 68.3 | 68.2 | 48.5 | 17.9 | 57.3 | 70.2 | 88.8 | 44.8 | 59.2 | 63.0(+10.0) |
| *Model Architecture: CLIP ViT-L/14* | | | | | | | | | | | | | |
| CLIP | 400M | 92.6 | 94.9 | 77.0 | 66.8 | 76.5 | 30.7 | 54.4 | 93.2 | 93.9 | 78.1 | 74.5 | 75.7 |
| LLM2CLIP | 3M | 64.8 | 95.4 | 72.9 | 66.4 | 18.8 | 10.4 | 54.8 | 47.3 | 88.3 | 26.8 | 52.8 | 54.4 |
| ProCLIP | 3M | 83.4 | 96.6 | 78.3 | 72.4 | 45.1 | 16.2 | 59.6 | 65.9 | 92.3 | 41.8 | 62.5 | 64.9 (+10.5) |
| LLM2CLIP | 15M | 70.1 | 95.2 | 72.3 | 66.4 | 32.4 | 9.5 | 58.0 | 54.3 | 88.3 | 26.6 | 54.0 | 57.0 |
| ProCLIP | 15M | 87.1 | 95.4 | 77.6 | 72.3 | 59.8 | 21.1 | 62.1 | 77.0 | 92.4 | 48.8 | 66.0 | 69.3 (+12.3) |
| LLM2CLIP | 30M | 71.2 | 94.0 | 70.5 | 67.0 | 32.1 | 11.3 | 57.8 | 54.7 | 89.3 | 28.8 | 56.4 | 57.5 |
| ProCLIP | 30M | 88.9 | 94.1 | 77.7 | 72.5 | 61.1 | 25.2 | 62.8 | 81.5 | 92.9 | 57.2 | 67.8 | 71.0 (+13.5) |
| *Model Architecture: EVA02-CLIP ViT-L/14* | | | | | | | | | | | | | |
| EVA02-CLIP | 2B | 92.9 | 98.8 | 89.8 | 73.8 | 88.8 | 35.1 | 60.6 | 93.7 | 95.1 | 76.3 | 78.2 | 80.3 |
| LLM2CLIP | 3M | 64.1 | 96.5 | 82.6 | 68.0 | 29.2 | 9.0 | 59.4 | 48.5 | 89.8 | 28.6 | 56.4 | 57.5 |
| ProCLIP | 3M | 82.7 | 97.9 | 88.4 | 73.6 | 57.6 | 16.5 | 63.5 | 67.6 | 93.8 | 45.4 | 66.8 | 68.5 (+11.0) |

## 4.2 MAIN RESULTS

**Cross-Modal Retrieval.** As shown in Tab. 1, ProCLIP consistently surpasses LLM2CLIP in both short- and long-text retrieval tasks across various datasets and model scales. On short-text datasets such as Flickr30k and COCO, ProCLIP achieves significant improvements in both image-to-text (I2T) and text-to-image (T2I) retrieval. For instance, with ViT-L/14 and 30M training samples, it reaches 95.0% I2T Recall@1 on Flickr30k—nearly 2 percentage points higher than LLM2CLIP. On long-text benchmarks including DOCCI, DCI, and Urban-1k, ProCLIP also exhibits clear advantages. Under ViT-B/16 trained on 30M samples, it attains 73.6% (I2T) and 76.9% (T2I) on DCI. Moreover, across all data scales from 3M to 30M, ProCLIP delivers stable gains, with particularly strong improvements in T2I retrieval. These results confirm that ProCLIP enhances performance in both short- and long-text scenarios.

**multilingual cross-modal Retrieval.** Benefiting from the LLM-based embedder, ProCLIP facilitates multilingual capabilities. As illustrated in Fig. 3, we compare the cross-lingual retrieval performance between LLM2CLIP and ProCLIP on the XM3600 benchmark (Thapliyal et al., 2022).

Experiment results demonstrate that our approach achieves superior multilingual performance. This enhancement is attributed to the improved alignment between the CLIP image encoder and the LLM-based embedder.

**Zero-Shot Classification.** In Tab. 2, we present the zero-shot classification performance on 11 downstream tasks across different data and model scales. We observe that LLM2CLIP significantly compromises the original generalization ability of CLIP. Even when utilizing 30M data points, compared to CLIP's ViT-B/32, ViT-B/16, and ViT-L/14, the average performance declines by 16.2%, 15.3%, and 18.2%, respectively. Compared to LLM2CLIP, our proposed ProCLIP method achieves significant performance improvements across all experimental conditions. Particularly, with a dataset of 30M samples, ProCLIP enhances the average performance by approximately 10%-13.5%. This notable improvement is primarily attributed to two factors: 1) The representation inheritance process allows the LLM embedder to inherit some knowledge from the original CLIP text encoder. 2) During the contrastive tuning phase, the introduction of a distillation loss as regularization helps to mitigate the forgetting of knowledge throughout the learning process.

**Robustness.** To evaluate the robustness of ProCLIP we report its performance across varying data sizes and model scales in Tab. 3. ProCLIP consistently achieves average improvements of 5.9%-9.3%. Notably, on challenging out-of-distribution datasets like ImageNet-A and ImageNet-R, ProCLIP outperforms LLM2CLIP by over 10 percentage points, highlighting its enhanced ability to handle distribution shifts and complex perturbations. These results demonstrate that ProCLIP not only improves retrieval and classification performance but also delivers robust and reliable results across diverse robustness scenarios, indicating substantial progress in generalization and resilience.

Table 3: Robustness performance. The best results are marked in **bold**.

| Method | Data | Robustness | | | | |
| --- | --- | --- | --- | --- | --- | --- |
| | | IN-V2 | IN-A | IN-O | IN-R | IN-S |
| *Model Architecture: CLIP ViT-L/14* | | | | | | |
| CLIP | 400M | 69.8 | 70.8 | 32.2 | 87.8 | 59.6 |
| LLM2CLIP | 3M | 49.0 | 46.6 | 32.4 | 75.0 | 44.8 |
| ProCLIP | 3M | **58.3** | **63.3** | 31.6 | **84.0** | **52.3** |
| LLM2CLIP | 15M | 50.8 | 50.1 | 33.8 | 78.2 | 46.3 |
| ProCLIP | 15M | **62.1** | **66.4** | 34.2 | **86.4** | **55.3** |
| LLM2CLIP | 30M | 52.7 | 52.7 | 34.0 | 78.6 | 47.3 |
| ProCLIP | 30M | **63.4** | **68.0** | 34.1 | **86.8** | **55.7** |
| *Model Architecture: EVA02-CLIP ViT-L/14* | | | | | | |
| EVA02-CLIP | 2B | 72.6 | 76.4 | 29.6 | 92.7 | 67.9 |
| LLM2CLIP | 3M | 51.8 | 50.4 | 28.8 | 79.1 | 50.6 |
| ProCLIP | 3M | **62.0** | **66.5** | 29.3 | **89.4** | **59.9** |

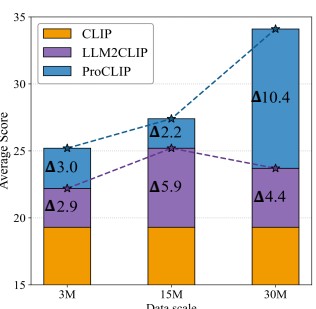

Figure 4: MMVP performance comparison. ProCLIP presents excellent performance.

Table 4: Comparison with other methods across different model scales and LLM embedders.

| Method | ViT Init | LLM Embedder | Data | ImageNet | COCO | | Flickr30k | |
| --- | --- | --- | --- | --- | --- | --- | --- | --- |
| | | | | | I2T | T2I | I2T | T2I |
| FLAME | random | Mistral-Nemo | 3M | 36.0 | 43.3 | 28.6 | 67.3 | 53.6 |
| ShareLock | DINOv2 B/14 | Llama3 | 3M | 52.1 | - | - | - | - |
| LIFT | random | NV-Embedv2 | 512M | 43.6 | 34.6 | 36.0 | 69.1 | 72.9 |
| LiT | CLIP B/16 | Llama3-CC | 3M | 51.0 | 56.2 | 41.9 | 85.2 | 71.9 |
| LLM2CLIP | CLIP B/16 | Llama3-CC | 3M | 45.8 | 60.5 | 44.8 | 88.0 | 75.3 |
| ProCLIP | CLIP B/16 | Llama3-CC | 3M | **54.8** | **61.7** | **46.8** | **89.4** | **77.6** |
| SAIL | DINOv2 L/14 | NV-Embedv2 | 3M | 54.0 | 45.4 | 32.9 | - | - |
| LiT | CLIP L/14 | Llama3-CC | 3M | 60.1 | 59.4 | 44.6 | 88.0 | 74.7 |
| LLM2CLIP | CLIP L/14 | Llama3-CC | 3M | 52.8 | 65.5 | 49.7 | 92.4 | 80.1 |
| ProCLIP | CLIP L/14 | NV-Embedv2 | 3M | 61.4 | 64.8 | 51.7 | 91.9 | **81.4** |
| ProCLIP | CLIP L/14 | Llama3-CC | 3M | **62.5** | **66.4** | 51.9 | **92.8** | 81.1 |

**Fine-Grained Understanding.** Fig. 4 presents the fine-grained vision-language understanding performance on the MMVP benchmark (Tong et al., 2024) using CLIP ViT-L/14. LLM2CLIP improves over CLIP by 2.9%, 5.9%, and 4.4% at 3M, 15M, and 30M data scales, respectively. Our ProCLIP model further advances these results, achieving gains of 3.0%, 2.2%, and 10.4% on the corresponding data scales. These improvements demonstrate that the LLM-based embedder enhances fine-grained semantic discrimination, and the consistent superiority of our method underscores the effectiveness of the progressive alignment strategy.

**Comparison with Other Methods.** To further prove the effectiveness of ProCLIP, we provide a comprehensive comparison of all recent LLM embedder-based CLIP models, including FLAME,

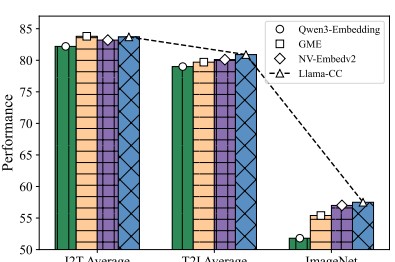

Figure 5: Ablation on different LLM-based embedders.

Table 5: Ablation on different components.

| Method | Stage 1 | | Stage 2 | | IN-1k | I2T Avg | T2I Avg |
| | $\mathcal{L}_{ins}$ | $\mathcal{L}_{struct}$ | $\mathcal{L}_{info}$ | $\mathcal{L}_{reg}$ | | | |
|---|---|---|---|---|---|---|---|
| CLIP | | | | | 74.5 | 66.9 | 55.5 |
| LLM2CLIP | | | | | 52.8 | 82.0 | 79.1 |
| ProCLIP | ✓ | | | | 58.9 | 69.3 | 79.4 |
| | ✓ | ✓ | | | 59.5 | 70.3 | 61.2 |
| | ✓ | ✓ | ✓ | | 59.2 | 82.9 | 80.2 |
| | ✓ | ✓ | ✓ | ✓ | **62.5** | **83.0** | **80.5** |

ShareLock, LIFT, SAIL, LiT, and our baseline LLM2CLIP. As shown in Tab. 4, under the same or lower training costs, ProCLIP consistently achieves superior performance across various model sizes. Benefiting from representation inheritance and self-distillation regularization, ProCLIP not only achieves significant performance improvements in In1k classification but also enhances general retrieval capabilities on COCO and Flickr30k.

### 4.3 Ablation Study

**Ablation of Different LLM-based Embedder.** As shown in Fig. 5, we compare different LLM embedders, including Qwen3-Embedding (8B), GME (7B), NV-Embedv2 (7B) , and Llama3-CC (8B) based on ViT-L/14 with 15M data. Llama-CC achieves the strongest overall performance in both ImageNet zero-shot classification and retrieval tasks. Notably, while different embedders show only minor variations in retrieval performance, they exhibit substantial differences in ImageNet classification accuracy. This suggests that the alignment discrepancy between each LLM embedder's feature space and the original CLIP space varies considerably, resulting in different degrees of degradation in general capabilities after image-text alignment.

**Ablation of Different Components.** To further validate the effectiveness of the methods proposed in this paper, we conduct a comprehensive ablation study on various components, as detailed in Tab. 5. Applying instance semantic distillation achieves 58.9% zero-shot accuracy on ImageNet-1k using only text data, indicating successful transfer of CLIP's textual representation capability to the MLP head. Incorporating the structural alignment loss further improves both classification and retrieval performance by enabling the LLM embedder to capture the global structural geometry of CLIP's text representation space, beyond point-wise semantic correspondences. After that, image-text contrastive learning significantly boosts retrieval performance but reduces ImageNet-1k accuracy due to image encoder overfitting. Introducing self-distillation mitigates this issue, improving classification accuracy from 59.2% to 62.5% while slightly reducing retrieval gains. Finally, applying structured self-distillation enhances both tasks by stabilizing the image representation space during fine-tuning, preventing excessive overfitting while preserving pretrained knowledge.

## 5 Conclusion

In this paper, we propose **ProCLIP**, a simple yet effective progressive vision-language alignment framework designed to improve the alignment when integrating the CLIP image encoder with an LLM-based embedder. The framework employs a curriculum learning–inspired progressive training strategy: it first aligns the LLM-based embedder's representation space with the original CLIP text encoder through knowledge distillation, effectively transferring pretrained semantic knowledge. Subsequently, it performs cross-modal alignment between the CLIP image encoder and the LLM-based embedder using image-text contrastive learning regularized by self-distillation to prevent overfitting and preserve pretrained knowledge. To ensure feature-space consistency, a complementary distillation strategy—comprising instance semantic and embedding structure alignment losses—is applied during text distillation and image self-distillation, respectively. Comprehensive experiments across varying data scales and model architectures demonstrate the effectiveness and generality of ProCLIP. We hope that our work offers valuable insights for advancing vision-language alignment. We will release all model weights and code to ensure full reproducibility.

## 6 ETHICS STATEMENT

This paper follows ICLR ethical guidelines and promotes responsible research practices. All experiments use publicly available datasets containing no personally identifiable, sensitive, or harmful content. The study involves no human subjects or vulnerable groups. We have evaluated potential societal impacts, including misuse risks, and conclude that our contributions advance scientific understanding without foreseeable harm.

## 7 REPRODUCIBILITY STATEMENT

We ensure full reproducibility by releasing all code and data in an anonymous repository. The paper details experimental configurations, including training procedures, model architectures, and hardware specifications. All benchmarks are publicly accessible, enabling consistent evaluation. These measures support verification and foster further research advancements.

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

# A  SUPPLEMENTARY MATERIAL

## A.1  THE USE OF LARGE LANGUAGE MODELS (LLMS)

We affirm that this paper is prepared and written entirely by us. We did not use any Large Language Models (LLMs) to generate the abstract, content, or any part of the text. The ideas, analysis, and conclusions presented are the sole product of the original thought and research. We employed only standard writing assistance tools, such as grammar checkers, for minor stylistic refinements.

## A.2  TRAINING DETAILS

Details of the hyperparameter configurations used for two-stage training of ProCLIP are presented in Tab. 6.

Table 6: Detailed hyperparameters for training ProCLIP.

| Hyperparameters of stage1 | | Hyperparameters of stage2 | |
|---|---|---|---|
| Batch size | 1024 (8 × 128) | Batch size | 4096 (8 × 512) |
| Optimizer | AdamW | Optimizer | AdamW |
| Weight decay | 0.05 | Weight decay | 0.05 |
| Adam $\beta$ | (0.9,0.98) | Adam $\beta$ | (0.9,0.98) |
| Adam $\epsilon$ | 1e-5 | Adam $\epsilon$ | 1e-6 |
| Learning rate | 1e-5 | Learning rate | 1e-5 |
| Learning rate schedule | cosine decay | Learning rate schedule | cosine decay |
| Epochs | 4 | Ema $\alpha$ | 0.999 |
| Training GPUs | 8×H100 | $\lambda_{\text{reg}}$ | 0.0004 |
| | | Epochs | 4 |
| | | Training GPUs | 8×H100 |

## A.3  DETAILS OF BENCHMARKS.

**Zero-Shot Classification & Linear Probe.** Following the previous works (Yang et al., 2023; Gu et al., 2024), we evaluate the zero-shot classification and linear probe performance of the models on 11 datasets. The detailed information about these datasets and the prompt used in zero-shot classification are presented in Tab. 7 and Tab. 10.

Table 7: List of zero-shot datasets with the data distribution and evaluation metrics.

| Dataset | Classes | Train size | Test size | Evaluation metric |
|---|---|---|---|---|
| Food101 | 102 | 75,750 | 25,250 | accuracy |
| CIFAR10 | 10 | 50,000 | 10,000 | accuracy |
| CIFAR100 | 100 | 50,000 | 10,000 | accuracy |
| SUN397 | 397 | 19,850 | 19,850 | accuracy |
| Cars | 196 | 8,144 | 8,041 | accuracy |
| Aircraft | 100 | 6,667 | 3,333 | mean per class |
| DTD | 47 | 3,760 | 1,880 | accuracy |
| Pets | 37 | 3,680 | 3,669 | mean per class |
| Caltech101 | 101 | 3,000 | 5,677 | mean-per-class |
| Flowers | 102 | 2,040 | 6,149 | mean per class |
| ImageNet | 1000 | 1,281,167 | 50,000 | accuracy |

**Robustness.** We evaluated the robustness of our model on five out-of-distribution datasets, including ImageNet-v2 Recht et al. (2019), ImageNet-A Hendrycks et al. (2021b), ImageNet-O Hendrycks et al. (2021b), ImageNet-R Hendrycks et al. (2021a), and ImageNet-Sketch Wang et al. (2019).

**Cross-Modal Retrieval.** Following the previous works (Huang et al., 2024; Cao et al., 2025), we evaluate the cross-modal retrieval performance of the models on 6 datasets: Flickr30k (Plummer et al., 2015), COCO (Lin et al., 2014), ShareGPT4V (Chen et al., 2024), Urban-1k (Zhang et al., 2024), DOCCI (Onoe et al., 2024), and DCI (Urbanek et al., 2024). The details information about these dataset are present on Tab. 8.

Table 8: Zero-shot image-text retrieval evaluation settings.

| Dataset | Test Images | Evaluation Protocol | Text type |
|---------|-------------|---------------------|-----------|
| MSCOCO | 5,000 | Image-to-Text & Text-to-Image | short |
| Flickr30k | 1,000 | Image-to-Text & Text-to-Image | short |
| ShareGPT4V | 1000 | Image-to-Text & Text-to-Image | long |
| Urban-1k | 1000 | Image-to-Text & Text-to-Image | long |
| DOCCI | 5000 | Image-to-Text & Text-to-Image | long |
| DCI | 7805 | Image-to-Text & Text-to-Image | long |

**Multilingual Retrieval.**

We evaluated the multilingual capabilities of our model on XM3600 Thapliyal et al. (2022). XM3600 contains 3,600 images covering a total of 36 languages, including Arabic (ar), Bengali(bn), Chinese-Simplified (zh), Croatian (hr), Czech (cs), Danish (da), Dutch (nl), English (en),Farsi (fa), Filipino (fil), Finnish (fi), French (fr), German (de), Greek (el), Hebrew (he), Hindi (hi), Hungarian (hu), Indonesian (id), Italian (it), Japanese (ja), Korean (ko),Maori(mi), Norwegian (no), Persian (fa), Polish (pl), Portuguese (pt), Romanian (ro), Russian (ru), Spanish (es), Swedish (sv), Swahili(sw), Thai (th), Turkish (tr), Telugu (te), Ukrainian (uk), and Vietnamese (vi).

**Fine-Grained Understanding.** We evaluated the fine-grained understanding capability of the VLM on MMVP-VLM Tong et al. (2024). MMVP-VLM consists of 150 samples in total, testing 9 patterns:

- ⊘ **Orientation and Direction**: Questions about the direction something is facing or moving, such as the direction the dog or duck is facing, or the orientation of the school bus.
- 🔍 **Presence of Specific Features**: Questions that focus on the existence or non-existence of certain elements or features in the image.
- 🔁 **State and Condition**: Questions that pertain to the state or condition of an object, such as whether a flag is blowing in the wind or if the ground is wet.
- ↥ **Quantity and Count**: Questions about the number of objects or features present in the image.
- 📍 **Positional and Relational Context**: This aspect refers to the model's ability to understand the position and relationship of objects or elements within an image in relation to each other and their surroundings.
- 🎨 **Color and Appearance**: Questions regarding the color of certain objects or elements.
- ⚙ **Structural and Physical Characteristics**: This category involves the model's ability to identify and analyze the physical attributes and structural features of objects in an image.
- **A** **Text**: Questions related to text or symbols present in the image.
- 📷 **Viewpoint and Perspective**: Questions concerning the perspective from which the photo was taken.

A.4    MORE RESULTS.

**Liner Probe.** We conduct linear probe evaluations of the model on 11 datasets. As shown in 9, our method consistently achieves superior performance. This advantage stems from our progressive alignment framework, which stabilizes training through two-stage regularization that prevents overfitting in the vision encoder while preserving generalization capability.

Table 9: Linear Probe performance on 11 datasets.

| Method | Data | Food101 | CIFAR10 | CIFAR100 | SUN397 | Cars | Aircraft | DTD | Pets | Caltech101 | Flowers | ImageNet | Avg. |
|---|---|---|---|---|---|---|---|---|---|---|---|---|---|
| *Model Architecture: CLIP ViT-B/32* | | | | | | | | | | | | | |
| CLIP | 400M | 88.6 | 95.1 | 80.1 | 73.4 | 80.8 | 44.9 | 76.3 | 89.3 | 92.7 | 94.7 | 74.3 | 80.9 |
| LLM2CLIP | 3M | 87.9 | 95.7 | **83.1** | 74.1 | 78.0 | **44.9** | 77.7 | **90.4** | 92.4 | 94.6 | 74.2 | 81.2 |
| ProCLIP | 3M | **88.4** | **95.9** | **83.1** | **74.3** | **79.5** | 44.1 | **78.2** | 90.3 | **92.6** | **95.0** | **74.4** | **81.4** |
| LLM2CLIP | 15M | 87.7 | 95.7 | 82.7 | 74.0 | 77.5 | 44.2 | **78.3** | **90.2** | 92.5 | 94.4 | 74.2 | 81.0 |
| ProCLIP | 15M | **88.7** | **95.9** | 82.8 | **74.8** | **80.8** | **44.9** | 78.1 | **90.2** | **92.8** | **95.1** | **74.4** | **81.7** |
| LLM2CLIP | 30M | 87.6 | 95.9 | 83.0 | 74.1 | 76.3 | 43.5 | 77.6 | **90.1** | **92.8** | 93.8 | 74.3 | 80.8 |
| ProCLIP | 30M | **88.2** | **96.0** | **83.1** | **75.1** | **79.0** | **43.8** | **77.8** | 89.8 | 92.6 | **94.9** | **74.5** | **81.4** |
| *Model Architecture: CLIP ViT-B/16* | | | | | | | | | | | | | |
| CLIP | 400M | 92.7 | 96.0 | 82.5 | 75.7 | 85.9 | 52.8 | 78.9 | 93.1 | 93.9 | 96.4 | 79.6 | 84.4 |
| LLM2CLIP | 3M | 91.6 | **97.0** | 84.5 | 76.0 | 82.1 | 50.1 | 80.3 | 92.3 | 93.6 | 95.7 | 79.6 | 83.9 |
| ProCLIP | 3M | **92.8** | 96.8 | **84.6** | **76.4** | **85.6** | **52.0** | **80.6** | **94.2** | **94.2** | **97.0** | **79.7** | **84.8** |
| LLM2CLIP | 15M | 91.9 | **97.0** | **84.9** | 75.6 | 83.7 | 50.7 | 80.4 | 92.9 | 93.8 | 96.6 | 79.6 | 84.3 |
| ProCLIP | 15M | **92.6** | 96.7 | 84.3 | **76.6** | **85.6** | **51.4** | **80.8** | **93.6** | **94.3** | **96.7** | **79.8** | **84.8** |
| LLM2CLIP | 30M | 91.3 | 96.6 | 84.8 | 75.3 | 80.6 | 48.2 | 80.3 | 92.5 | 93.4 | 95.0 | 79.7 | 83.4 |
| ProCLIP | 30M | **92.3** | **96.6** | **85.7** | **77.0** | **84.7** | **50.1** | **81.2** | **93.1** | **94.0** | **96.7** | 79.5 | **84.6** |
| *Model Architecture: CLIP ViT-L/14* | | | | | | | | | | | | | |
| CLIP | 400M | 95.3 | 89.1 | 87.2 | 79.4 | 90.7 | 63.0 | 81.8 | 95.3 | 96.9 | 98.8 | 82.9 | 88.1 |
| LLM2CLIP | 3M | 94.5 | **98.6** | **89.2** | 79.6 | 86.7 | 57.7 | 83.4 | 94.1 | 96.4 | 97.1 | **82.5** | 87.2 |
| ProCLIP | 3M | **95.3** | 98.5 | 88.8 | **80.3** | **90.3** | **61.0** | **83.6** | **95.2** | **96.9** | **98.7** | 81.9 | **88.2** |
| LLM2CLIP | 15M | 94.4 | **98.5** | **88.8** | 78.5 | 86.0 | 55.0 | 82.7 | 93.9 | 95.9 | 97.1 | **82.6** | 86.7 |
| ProCLIP | 15M | **95.2** | 98.4 | 88.6 | **79.7** | **90.5** | **61.4** | **83.3** | **95.3** | **96.8** | **98.7** | **83.0** | 86.7 |
| LLM2CLIP | 30M | 94.1 | 98.2 | 88.4 | 78.7 | 84.6 | 54.8 | 82.4 | 93.7 | 95.8 | 96.5 | 82.2 | 86.3 |
| ProCLIP | 30M | **95.1** | **98.4** | **89.0** | **80.3** | **90.0** | **60.0** | **83.9** | **95.2** | **96.8** | **98.5** | **82.7** | **88.2** |
| *Model Architecture: EVA02-CLIP ViT-L/14* | | | | | | | | | | | | | |
| EVA02-CLIP | 2B | 95.6 | 99.5 | 94.2 | 80.4 | 94.2 | 69.5 | 85.0 | 94.8 | 97.6 | 99.4 | 84.1 | 87.4 |
| LLM2CLIP | 3M | 94.1 | **99.5** | 93.3 | 79.4 | 85.0 | 54.3 | 84.0 | 93.2 | 97.3 | 96.9 | 84.1 | 87.4 |
| ProCLIP | 3M | **95.3** | **99.5** | **94.0** | **81.0** | **93.9** | **65.7** | **85.9** | **95.4** | **97.8** | **99.3** | **84.5** | **90.2** |

Table 10: Full list of prompts to evaluate the performance of zero-shot classification on 11 visual recognition datasets.

**CIFAR 10 & CIFAR 100**
a photo of a {label}.
a high contrast photo of a {label}.
a photo of a big {label}.
a low contrast photo of the {label}.
a photo of the small {label}.
a blurry photo of a {label}.
a bad photo of a {label}.
a photo of the {label}.
a high contrast photo of the {label}.
a photo of the big {label}.
a black and white photo of a {label}.
a good photo of a {label}.
a blurry photo of the {label}.
a bad photo of the {label}.
a low contrast photo of a {label}.
a photo of a small {label}.
a black and white photo of the {label}.
a good photo of the {label}.

**Food101**
a photo of {label}, a type of food.

**Caltech101**
a photo of a {label}.
a sketch of a {label}.
a embroidered {label}.
an origami {label}.
a sculpture of a {label}.
a rendition of the {label}.
the plushie {label}.
a drawing of the {label}.
a painting of a {label}.
a tattoo of a {label}.
a cartoon {label}.
art of a {label}.
a photo of the {label}.
a sketch of the {label}.
the embroidered {label}.
the origami {label}.
a doodle of the {label}.
a plastic {label}.
a toy {label}.
a {label} in a video game.
graffiti of a {label}.
a painting of the {label}.
a tattoo of the {label}.
the cartoon {label}.
art of the {label}.
a sculpture of a {label}.
a rendition of a {label}.
a plushie {label}.
a drawing of a {label}.
the plastic {label}.
the toy {label}.
the {label} in a video game.
graffiti of the {label}.

**Stanford Cars**
a photo of a {label}.
a photo of my dirty {label}.
a photo of the {label}.
a photo of my clean {label}.
a photo of my {label}.
a photo of my new {label}.
i love my {label}!
a photo of my old {label}.

**DTD**
a photo of a {label} texture.
a photo of the {label} texture.
a photo of a {label} pattern.
a photo of the {label} pattern.
a photo of a {label} thing.
a photo of the {label} thing.
a photo of a {label} object.
a photo of the {label} object.

**FGVC Aircraft**
a photo of a {label}, a type of aircraft.
a photo of the {label}, a type of aircraft.

**Flowers102**
a photo of a {label}, a type of flower.

**Pets**
a photo of a {label}, a type of pet.

**SUN39**
a photo of a {label}.
a photo of the {label}.

**ImageNet**
a bad photo of a {label}.
a low resolution photo of the {label}.
a cropped photo of the {label}.
a bright photo of a {label}.
a drawing of a {label}.
a close-up photo of a {label}.
a pixelated photo of the {label}.
a plastic {label}.
a photo of the {label}.
a photo of one {label}.
the origami {label}.
an origami {label}.
a photo of the clean {label}.
a photo of a weird {label}.
a sketch of the {label}.
a jpeg corrupted photo of the {label}.
a photo of the small {label}.
a drawing of the {label}.
a dark photo of a {label}.
itap of my {label}.
a photo of many {label}.
a rendering of a {label}.
a tattoo of a {label}.
a photo of a clean {label}.
a photo of my {label}.
a black and white photo of the {label}.
a sculpture of the {label}.
a photo of the dirty {label}.
a good photo of the {label}.
a doodle of a {label}.
the {label} in a video game.
a low resolution photo of a {label}.
a photo of a large {label}.
a blurry photo of a {label}.
a embroidered {label}.
a good photo of a {label}.
a photo of the weird {label}.
a photo of the large {label}.
itap of a {label}.
a photo of a cool {label}.
a sculpture of a {label}.
graffiti of a {label}.
the embroidered {label}.
a photo of a dirty {label}.
the plastic {label}.
a painting of the {label}.
a bright photo of the {label}.
a jpeg corrupted photo of a {label}.
a rendering of the {label}.
a close-up photo of the {label}.
a sketch of a {label}.
the toy {label}.
a rendition of a {label}.
a cartoon {label}.
a pixelated photo of a {label}.
a plushie {label}.
the cartoon {label}.
a black and white photo of a {label}.
graffiti of the {label}.
a photo of a small {label}.
a photo of the hard to see {label}.
a bad photo of the {label}.
a photo of a hard to see {label}.
a dark photo of the {label}.
a photo of the cool {label}.
a painting of a {label}.
a cropped photo of a {label}.
a blurry photo of the {label}.
a {label} in a video game.
a photo of a {label}.
a doodle of the {label}.
a rendition of the {label}.
a photo of a nice {label}.
art of a {label}.
itap of the {label}.
a photo of the nice {label}.
art of the {label}.
the plushie {label}.
a toy {label}.
a tattoo of the {label}.

