# OpenReview forum: "Progressive Vision-Language Alignment via LLM-based Embedder"
_ICLR.cc/2026/Conference — ICLR 2026 Conference Withdrawn Submission_

### Official Review · Reviewer_Ur8h · 2025-10-30

**Soundness:** 3
**Presentation:** 3
**Contribution:** 2
**Rating:** 4
**Confidence:** 3

**Summary:**

This paper addresses the problem of integrating a large language model (LLM)–based text embedder into an existing CLIP image encoder without losing the generalization learned by CLIP’s original text encoder. The authors argue that directly aligning the CLIP image encoder with an LLM embedder via contrastive learning risks discarding CLIP’s pretrained knowledge and causing overfitting. Empirically, ProCLIP improves zero-shot classification by ~6.8%–13.5% over the baseline LLM2CLIP across settings, shows consistent retrieval gains, better multilingual retrieval, and improved robustness and fine-grained V-L understanding. The paper includes several ablations showing the contribution of the two-stage design and the different loss terms.

**Strengths:**

1. the two-stage curriculum (distill CLIP text → align with LLM embeddings → constrained contrastive tuning) is a natural and effective strategy to avoid “from-scratch” alignment and catastrophic forgetting of CLIP priors.
2. experiments cover multiple model scales, data scales, tasks (zero-shot classification, retrieval, multilingual retrieval, robustness, fine-grained V-L), and ablations (different LLM embedders, loss components). Results show consistent improvements and practical applicability.

**Weaknesses:**

1. computing and storing embeddings for tens of millions of captions (3M–30M) is costly in storage and IO; no discussion or evaluation is provided about how sensitive performance is to embedding staleness, prompt templates, or the choice to do offline vs. online embedding.

2. some hyperparameter values are inconsistent between sections (e.g., weight decay reported as 0.2 in Sec.4.1 vs 0.05 in Appendix Table 6).

3. Training time, GPU-hours, and storage requirements are not reported.

**Questions:**

1. Please clarify precisely which components are trainable and which are frozen in each stage (stage 1: MLP only or other modules? stage 2: is the CLIP image encoder fine-tuned and is the LLM kept frozen?). Also specify the MLP architecture (depth, hidden dims) used in all experiments.

2. How are LLM embeddings generated (prompt templates, normalization, use of special tokens)? What is the storage footprint for precomputing embeddings at 30M captions? Have you tried online embedding (computing LLM outputs on the fly) or incremental updates when the LLM changes — how sensitive is ProCLIP to embedding “staleness” or to different prompt choices?

---

### Official Review · Reviewer_sNiT · 2025-10-30

**Soundness:** 3
**Presentation:** 2
**Contribution:** 2
**Rating:** 4
**Confidence:** 5

**Summary:**

This paper introduces a progressive alignment approach to replace CLIP’s weak text encoder with a stronger LLM-based encoder, enabling multilingual and long-text multimodal understanding. The method employs a two-stage alignment process and demonstrates superior performance over direct alignment methods without knowledge distillation (LLM2CLIP), achieving consistent improvements across cross-modal retrieval benchmarks.

**Strengths:**

The proposed method achieves clear improvements over prior work (LLM2CLIP). Ablation studies validate the effectiveness of the progressive alignment strategy and knowledge distillation, showing consistent performance gains across various retrieval tasks.

**Weaknesses:**

The proposed method appears somewhat ad-hoc and conceptually straightforward. Its only addition over LLM2CLIP is the knowledge distillation stage, which lacks substantial novelty.

Although the method improves retrieval performance, the degradation in zero-shot classification is unexpected, as these two tasks are typically correlated. Prior work [1] demonstrated improvements in both, whereas ProCLIP and LLM2CLIP show weaker alignment. More analysis and discussion are needed to explain this discrepancy.

The ablation study is overly broad; it would be valuable to show which components contribute most to specific tasks. Furthermore, results and discussion on compositional retrieval—an especially relevant setting—would strengthen the paper’s evaluation.

Figures 1 and 2 have significant overlap, and Figure 2 includes excessive, repetitive information that could be streamlined.



[1] Zhang L, Yang Q, Agrawal A. Assessing and Learning Alignment of Unimodal Vision and Language Models[C]//Proceedings of the Computer Vision and Pattern Recognition Conference. 2025: 14604-14614.

**Questions:**

Why does zero-shot classification performance degrade while retrieval performance improves? Which part result such behaviour? How is it different from [1] ?t

 [1] Zhang L, Yang Q, Agrawal A. Assessing and Learning Alignment of Unimodal Vision and Language Models[C]//Proceedings of the Computer Vision and Pattern Recognition Conference. 2025: 14604-14614.

---

### Official Review · Reviewer_2RyM · 2025-10-30

**Soundness:** 2
**Presentation:** 3
**Contribution:** 2
**Rating:** 4
**Confidence:** 3

**Summary:**

This paper introduces ProCLIP, a progressive vision-language alignment framework leveraging LLM-based text embedders to address CLIP’s limitations in input length, multilinguality, and fine-grained semantics. The method uses a two-stage training process: knowledge distillation from CLIP’s text encoder to the LLM embedder, followed by further alignment of the image encoder with the LLM embedder via contrastive learning and self-distillation. The authors report improvements in zero-shot classification and cross-modal retrieval tasks.

**Strengths:**

1. Integration: The progressive alignment strategy is a well-designed combination of existing techniques.
2. Clarity: The methodology and experiments are clearly presented.
3. Practical value: The approach addresses real limitations of CLIP in handling long and multilingual text.

**Weaknesses:**

1. Limited experimental scope: Experiments are only conducted on retrieval and classification. There is no evaluation on other important vision-language tasks (e.g., VQA, captioning, visual reasoning), which limits the evidence for general applicability.
2. Decreased robustness and visual understanding: According to Table 2 and Table 3, ProCLIP reduces visual understanding capability and robustness compared to the original CLIP, undermining the claim of preserving generalization.
3. Incremental novelty: The core ideas are closely related to recent works (FLAME, LLM2CLIP, LiT), and the main novelty is in the integration rather than new algorithms.
3. Lack of failure analysis: The paper does not discuss scenarios where ProCLIP underperforms or the reasons for decreased robustness.

**Questions:**

1. Can the authors provide experiments on a broader set of vision-language tasks (e.g., VQA, image captioning, visual reasoning) to better demonstrate generalization?
2. What are the underlying reasons for the observed decrease in visual understanding and robustness? Are there ways to mitigate this trade-off?
3. Would combining ProCLIP with other alignment paradigms (e.g., RLHF, instruction tuning) help address the robustness issue?
4. Can the authors discuss practical deployment costs and limitations more explicitly?

---

### Official Review · Reviewer_Uxeu · 2025-11-01

**Soundness:** 2
**Presentation:** 2
**Contribution:** 2
**Rating:** 4
**Confidence:** 4

**Summary:**

This paper addresses a timely and important problem: how to effectively integrate the rich, open-world knowledge of Large Language Models (LLMs) into the CLIP framework to overcome its limitations in handling long texts, multilingual inputs, and fine-grained semantics. The authors propose ProCLIP, a two-stage, curriculum learning-based alignment framework. The first stage (“Representation Inheritance”) distills knowledge from the original CLIP text encoder into the LLM-based embedder. The second stage (“Contrastive Tuning”) performs image-text contrastive learning, regularized by a self-distillation constraint on the image encoder to prevent catastrophic forgetting. The paper presents extensive experiments across various tasks (zero-shot classification, cross-modal retrieval, multilingual retrieval, robustness) and data scales, demonstrating significant improvements over the primary baseline, LLM2CLIP.

**Strengths:**

1. The paper correctly identifies a key limitation in existing LLM-augmented CLIP methods: direct contrastive alignment harms the pretrained CLIP’s generalization.
2. The proposed two-stage framework is well-designed and leverages established techniques (knowledge distillation, self-distillation) in a principled manner. The use of both instance-level and structure-level alignment losses is a thoughtful detail.
3. The evaluation is thorough, covering a wide range of tasks, data scales (3M, 15M, 30M), and model architectures (ViT-B/32, B/16, L/14, EVA02-CLIP).
4. ProCLIP demonstrates clear and consistent gains over LLM2CLIP, especially in zero-shot classification and robustness. The performance on long-text and multilingual retrieval is also solid.

**Weaknesses:**

1. Although the progressive alignment idea is practical, it mainly combines existing techniques (distillation + contrastive learning + self-distillation) rather than introducing a fundamentally new principle. The novelty is incremental rather than conceptual.
2. No theoretical justification or deeper analysis is provided on why progressive alignment preserves generalization. The role of structure alignment vs. self-distillation is only empirically shown.
3. The core motivation is to replace CLIP’s text encoder, yet Stage 1 critically depends on it to generate distillation targets (Eq. 2–3). This creates a logical loop: the LLM embedder inherits knowledge from a teacher that is itself limited to 77 tokens and English.
4. The primary baseline, LLM2CLIP, is not fairly compared: ProCLIP uses batch=4096 and 8×H100 GPUs (Appendix A.2), while LLM2CLIP’s training details (batch size, epochs, regularization) are not disclosed. The performance gap may stem from training engineering, not methodological superiority.
5. The paper claims gains in multilingual and long-text understanding but fails to compare with SigLIP 2 [Tschannen et al., arXiv 2025] (multilingual VLM) and LoTLIP [Wu et al., NeurIPS 2024] (long-text optimized). The omission of these baselines significantly weakens the paper’s empirical contribution and makes it difficult to assess the true state-of-the-art performance of the proposed method.
6. The most crucial ablation—ProCLIP without Stage 1 (i.e., direct contrastive tuning with the same Stage 2 protocol)—is missing. Without it, the necessity of the two-stage design cannot be established.
7. In Table 5, adding the L_struct loss in Stage 1 causes a dramatic drop in T2I retrieval performance (from 79.4 to 61.2). However, the text claims that L_struct “further improves both classification and retrieval performance.” This is a direct contradiction that needs to be addressed and explained. The paper also fails to explain why Stage 1 with only L_ins already achieves near-optimal T2I performance (79.4%), suggesting L_struct may be unnecessary or even harmful.
8. Despite improvements over LLM2CLIP, ProCLIP still underperforms the original CLIP (400M) on zero-shot classification (Table 2: 71.0% vs. 75.7% with ViT-L/14). This indicates that LLM integration cannot compensate for the lack of large-scale pretraining, questioning the method’s fundamental value.
9. The claim that LLMs provide “richer open-world knowledge” is never validated: all evaluations are on closed-set tasks. No experiments test open-vocabulary generalization, compositional reasoning, or novel concept understanding.
10. Performance is highly sensitive to the LLM choice (Fig. 5): Qwen3 drops ImageNet accuracy by >6% vs. LLaMA3-CC. The paper offers no analysis of why certain LLMs align better, raising concerns about reliability.
11. The paper lacks discussion on computational overhead. Two-stage training doubles optimization cost, and no runtime or memory analysis is given. The requirement of 8×H100 GPUs and batch=4096 makes the method inaccessible to most academic labs, contradicting the “simple yet effective” framing and limiting real-world adoption.

**Questions:**

My major concerns are outlined in the "Weaknesses" part.

---

### Note · Authors · 2025-11-25

I have read and agree with the venue's withdrawal policy on behalf of myself and my co-authors.